# STRUCTDROP: A STRUCTURED RANDOM ALGORITHM TOWARDS EFFICIENT LARGE-SCALE GRAPH TRAINING

## ABSTRACT

Training GNNs over large graphs is a long-standing challenge due to the inefficiency of the message passing mechanism. Message passing, typically represented as the production between sparse adjacency matrix and node features, is difficult to be accelerated with commodity hardware, such as GPUs. Prior dropping based mechanism (e.g., edge or node dropping), can be adopted to reduce the computation cost of sparse matrix multiplication. However, two under-explored pain points still persist in this paradigm: ① *Inefficiency.* Dropping-based methods lack hardware efficiency. Such mechanism randomly remove non-zero entries from edge indices, which later needs to be converted into sparse matrix format for computational ease. This conversion may counteract the speedup gained from reducing FLOPs. ② *Poor generalization.* Previous sampling-based method utilizes a fixed subset of nodes or edges to emphasize on efficiency, but sacrifice generalizability due to under-fitting on the remaining subgraph. Aiming to promote the accuracy-efficiency trade-off, we propose Structured Dropout, a.k.a, `StructDrop`. Specifically, we remove a set of selected columns directly from the sparse adjacency matrix format, hence bypassing the sparse matrix reconstruction and data access. To further mitigate the training shifting due to random column-row pair dropping, we adopt instance normalization following the sparse production. Comprehensive experiments on four benchmark datasets and four popular GNNs validate the superiority of our framework: `StructDrop` achieves up to 5.29x end-to-end speedup with negligible accuracy loss or even better accuracy compared with vanilla GNNs.

## 1 INTRODUCTION

Graph Neural Networks (GNNs) have made significant advancements in various graph-related tasks Hamilton et al. (2017a); Hu et al. (2020); Ying et al. (2018); Jiang et al. (2022); Zhou et al. (2022; 2023). Specifically, GNNs process the underlying graph structure and node features in a layer-wise manner with two interleaved phases: aggregation and update. During the aggregation phase, each node accumulates messages from its direct neighbors, which is computationally realized by sparse matrix-based operations to multiply the set of node features with a sparse adjacency matrix. Following this, in the update phase, nodes transform the aggregated features with a differentiable layer (e.g., multi-layer perceptron) dominated by dense matrix-based operations.

Despite their strong performance, training GNNs is time-inefficient, especially on large graphs. As shown in Figure 1, we analyze the fine-grained time cost of GNNs where `SpMM` and `MatMul` represents the sparse and dense operators, respectively. Notably, the neighborhood aggregations included at forward and backward propagations consume 70-90% of the total GNN training time, as supported by Han et al. (2023). This inefficiency stems from the nature of sparse matrix operations, which require numerous random memory accesses with minimal data reuse. Several works have highlighted that community hardware (e.g., CPUs and GPUs) designed on the single-instruction multiple-data (SIMD) principle will struggle in efficiently accessing neighborhood features with discontinuous indexes Duan et al. (2022); Han et al. (2016); Liu et al. (2023b).

Existing work towards reducing the time cost of neighborhood aggregation mainly adopt randomized dropping algorithms, which can be roughly grouped into two categories. Firstly, edge-oriented

dropping methods Rong et al. (2019); Eppstein et al. (1997); Liu et al. (2023b) remove part of the edges randomly during training, or deterministically in preprocessing stage. Secondly, node-oriented dropping methods Feng et al. (2020); Chiang et al. (2019); Hamilton et al. (2017b) prune certain nodes and their associated edges from the input graph. However, from the efficiency aspect, an issue with both approaches is that the overhead from removing edges or nodes may counteract the speedup from the FLOPS reduction. Specifically, this is due to the need to reconstruct the sparse adjacency matrix after removing edges or nodes from the input graph, which involves processing the whole graph and is notably time-consuming.

A less explored method to speed up the aggregation phase is to use a fast but approximated version of the SpMM instead of the exact one. To illustrate, consider a linear operation involving two matrices, $\mathbf{A} \in \mathbb{R}^{n \times m}$ and $\mathbf{B} \in \mathbb{R}^{m \times q}$. We first create reduced matrices $\mathbf{A}' \in \mathbb{R}^{n \times k}$ and $\mathbf{B}' \in \mathbb{R}^{k \times q}$ ($k < m$) by choosing $k$ representative columns from $\mathbf{A}$ and their corresponding rows from $\mathbf{B}$, referred to as column-row pairs. This approximation, $\mathbf{AB} \approx \mathbf{A}'\mathbf{B}'$, aims to reduce both the number of floating-point operations (FLOPs) and the data that needs to be accessed, as only $k/m$ of the column-row pairs are processed. This method avoids the need to reconstruct a sparse matrix by structurally selecting entire columns and rows. Although this approach has shown promise in other fields Adelman et al. (2021), our tests reveal that it significantly reduces the accuracy of GNNs, leading to even a 8% loss in accuracy (as shown in Table 1) on standard datasets and models, which is impractical for real-world applications.

In this work, we promote the **accuracy-efficiency trade-off** via approximating the sparse matrix production in both the forward and backward processes of GNNs. Based on the column-row pair sampling, our core idea is to adapt the sampling policy and normalize the result of SpMM to stably approximate the neighbor aggregation. Specifically, prior research suggests the probability of choosing each column-row pair should be in proportion to the production of the respective row norm and column norm Drineas et al. (2006). Interestingly, we observed that the column-row pairs selected in the forward pass exhibited a remarkable consistency across nearby iterations. We hypothesize that this consistency will cause under-fitting problem as they only utilize the same subset of nodes and edges during training. Drawing from this insight, we propose a straightforward strategy: **the uniform selection of column-row pairs.** Namely, we assign the same probability to be sampled for each column-row pair and term such structured dropping as StructDrop. Surprisingly, we found that this simple strategy can often outperform the complicated norm-based one in the graph learning problem. To further reduce the negative impact of the variance from uniform sampling, we propose to utilize instance normalization following the approximated production to stabilize the training process. In summary, our contributions are summarized as follows:

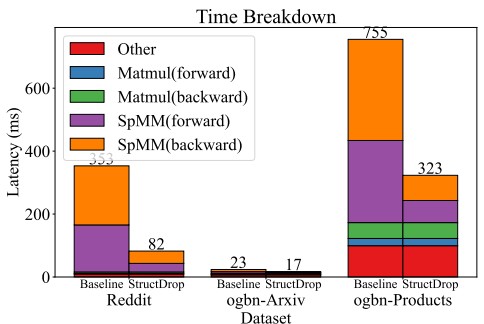

Figure 1: The time profiling of a three-layer GCNs on different datasets. SpMM may take 70∼90% of the total time. **Our method (StructDrop) can reduce the total training time by** $5.29\times$ as shown in table 2. We measure the time on a NVIDIA A40 GPU. The detailed software and hardware information can be found in Appendix A.

- We explore to speedup GNN training from a novel randomized dropping perspective. We approximate sparse matrix multiplication at forward and backward paths with sampling a subset of the column-row pairs to reduce FLOPs and data access with accuracy preserved.

- We propose a hybrid solution of random dropping and normalization to maintain generalizability with efficiency. We design a straightforward yet effective strategy, uniform sampling, which overcomes underfitting in global graph. Additionally, we recommend incorporating instance normalization into the sampling process so as to mitigate the embedding shift resulted from sampling.

- We conduct comprehensive experiments on seven popular GNNs and four large graphs. Compared with vanilla GNN, our achieve up to 5.29x speedup with negligible accuracy loss or better accuracy. We obtain a superior efficiency or accuracy while keeping the other metric comparable with other baselines.

## 2 PRELIMINARIES AND BACKGROUND

### 2.1 GRAPH NEURAL NETWORKS

We consider an undirected graph $G = (\mathcal{V}, \mathcal{E})$, where $\mathcal{V}$ and $\mathcal{E}$ denote the sets of nodes and edges, respectively, of size $N = |\mathcal{V}|$ and $E = |\mathcal{E}|$. Let $\boldsymbol{A} \in \mathbb{R}^{n \times n}$ denote the adjacency matrix, $\boldsymbol{A}_{i,j} = 1$ if $(v_i, v_j) \in \mathcal{E}$ else $\boldsymbol{A}_{i,j} = 0$, and let $\boldsymbol{X} \in \mathbb{R}^{n \times d}$ denotes the feature matrix. Based on the spatial message passing, GNNs learn the node representation through aggregating the neighbors' embeddings and combining with itself layer by layer. For example, the node embedding learning at the $l^{\text{th}}$ layer of Graph Convolutional Network (GCN) Kipf & Welling (2017) is defined as:

$$\boldsymbol{H}^{(l)} = \tilde{\boldsymbol{A}} \boldsymbol{X}^{(l-1)} \boldsymbol{W}^{(l)}, \boldsymbol{X}^{(l)} = \text{ReLU}(\boldsymbol{H}^{(l)}), \tag{1}$$

where $\boldsymbol{X}^{(l)} \in \mathbb{R}^{N \times d}$ is the node embedding matrix at the $l^{\text{th}}$ layer and $\boldsymbol{X}^{(0)} = \boldsymbol{X}$; $\tilde{\boldsymbol{A}} = \tilde{\boldsymbol{D}}^{-\frac{1}{2}}(\boldsymbol{A} + \boldsymbol{I})\tilde{\boldsymbol{D}}^{-\frac{1}{2}}$ is normalized adjacency matrix, $\tilde{\boldsymbol{D}}$ is the diagonal degree matrix of $\boldsymbol{A} + \boldsymbol{I}$; $\boldsymbol{W}^{(l)} \in \mathbb{R}^{d \times d}$ is trainable weight. In practice, $\tilde{\boldsymbol{A}}$ is often stored in sparse matrix format like compressed sparse row (CSR) to save the computation cost Fey & Lenssen (2019). Each training step has two phases, i.e., forward and backward passes. From the implementation perspective, its computation can be written as:

$$\text{Forward Pass} \quad \boldsymbol{J}^{(l)} = \texttt{MatMul}(\boldsymbol{X}^{(l-1)}, \boldsymbol{W}^{(l)}),$$
$$\boldsymbol{H}^{(l)} = \texttt{SpMM}(\tilde{\boldsymbol{A}}, \boldsymbol{J}^{(l)}), \tag{2a}$$
$$\text{Backward Pass} \quad \nabla \boldsymbol{J}^{(l)} = \texttt{SpMM}(\tilde{\boldsymbol{A}}^{\top}, \nabla \boldsymbol{H}^{(l)}), \tag{2b}$$
$$\nabla \boldsymbol{X}^{(l-1)} = \texttt{MatMul}(\nabla \boldsymbol{J}^{(l)}, \boldsymbol{W}^{(l)}),$$
$$\nabla \boldsymbol{W}^{(l)} = \texttt{MatMul}(\boldsymbol{X}^{(l-1)\top}, \nabla \boldsymbol{J}^{(l)}),$$

where $\texttt{SpMM}(\cdot, \cdot)$ is the Sparse-Dense Matrix Multiplication and $\texttt{MatMul}(\cdot, \cdot)$ is the normal Dense-Dense Matrix Multiplication. From above, we can see that **each training step has exactly two SpMM operations.** In practice, although using a sparse matrix format can reduce memory cost compared to using a dense representation of the adjacency matrix, it is still notoriously inefficient on commodity hardware due to the cache miss problem Han et al. (2016). As shown in Figure 1, we observed that $\texttt{SpMM}$ can take a large fraction of the training time.

### 2.2 FAST MATRIX MULTIPLICATION WITH SAMPLING

Given matrices $\boldsymbol{X} \in \mathbb{R}^{n \times m}$ and $\boldsymbol{Y} \in \mathbb{R}^{m \times q}$, our goal is to efficiently estimate the matrix product $\boldsymbol{XY}$. The Truncated Singular Value Decomposition (SVD) offers an optimal low-rank approximation of the product $\boldsymbol{XY}$ Adelman et al. (2021), but its computational cost is almost equivalent to matrix multiplication. To address the challenge, sampling algorithms have been introduced as a means of approximating the matrix product $\boldsymbol{XY}$. Such methods sample $k$ columns from $\boldsymbol{X}$ and the corresponding rows from $\boldsymbol{Y}$, resulting in smaller matrices. These matrices are then multiplied in the traditional manner Drineas et al. (2006). Such an approach cuts down the computational complexity from $\mathcal{O}(mnq)$ to $\mathcal{O}(knq)$. Mathematically, the approximation is given by:

$$\boldsymbol{XY} \approx \sum_{t=1}^{k} \frac{1}{s_t} \boldsymbol{X}_{:,i_t} \boldsymbol{Y}_{i_t,:} = \texttt{approx}(\boldsymbol{XY}), \tag{3}$$

where $\boldsymbol{X}_{:,i}$ and $\boldsymbol{Y}_{i,:}$ represent the $i^{\text{th}}$ column of $\boldsymbol{X}$ and the $i^{\text{th}}$ row of $\boldsymbol{Y}$, respectively. Within this context, we define the $(\boldsymbol{X}_{:,i}, \boldsymbol{Y}_{i,:})$ as the $i^{\text{th}}$ column-row pair. The term $k$ denotes the number of samples. $\{p_i\}_{i=1}^{m}$ represents a probability distribution across the column-row pairs. $i_t \in \{1, \cdots m\}$ is the index of the sampled column-row pair at the $t^{\text{th}}$ trial. $s_t$ is the scale factor. Drineas et al. (2006) indicates that setting $s_t = \frac{1}{kp_{i_t}}$ guarantees the expectation of low-rank approximation equals to the results of actual matrix multiplication. Furthermore, the approximation error is minimized when the sampling probabilities are proportional to the product of the norms of column-row pairs:

$$p_i = \frac{||\boldsymbol{X}_{:,i}||_2 \, ||\boldsymbol{Y}_{i,:}||_2}{\sum_{j=1}^{m} ||\boldsymbol{X}_{:,j}||_2 \, ||\boldsymbol{Y}_{j,:}||_2}. \tag{4}$$

Though the above sampling method effectively accelerates matrix multiplication Drineas et al. (2006), its direct application to neural networks might not be optimal. This is because it overlooks the unique distribution of neural network weights. Observations indicate that neural network weight distributions tend to remain centered around zero during training Glorot & Bengio (2010); Han et al. (2015). Using this insight, Adelman et al. (2021) introduced the **Top-$k$ sampling** method: deterministically selecting the $k$ column-row pairs that have the highest values according to Equation 4, without any scaling. This equates to setting the probability $p_i$ of the top $k$ column-row pairs to 1, and to 0 for the others, with the scale factor $s_{i_t}$ being consistently 1.

Furthermore, Liu et al. (2023a) adapted the top-k sampling technique to the domain of graph learning. To guarantee gradient unbiasedness, **they restricted the use of randomized matrix multiplication to the backward pass only, i.e., $\nabla J^{(l)} = \text{SpMM}(\tilde{A}^\top, \nabla H^{(l)})$ in Equation 2b**. This decision was influenced by the understanding that the non-linear activation functions can alter the expected outcome of the approximated matrix multiplication Liu et al. (2023a). While this approach preserves the final model accuracy, its potential for computational speedup is limited at $2\times$, given that it optimizes only the backward computations.

In the following sections, we investigate the feasibility to employ randomized matrix multiplication throughout the entire training process with better acceleration while effectively addressing the challenge of preserving accuracy.

## 3 METHODOLOGY

We propose `StructDrop` as an efficient yet accurate graph training scheme. We first present an interesting finding, that the sound theoretical guarantee of minimal error in Top-$k$ sampling might not be the most robust algorithm. We analyze and conduct experiments to answer why Top-$k$ sampling cannot maintain the accuracy in Sec 3.1. Based on this observation, we propose `StructDrop` in Section 3.2, which uniformly select the column-row pairs during graph training. In Sec 3.3, we further suggest integrating instance normalization to further enhance the stability of training process when working with sampling based scheme.

### 3.1 THE UNDER-FITTING PROBLEM IN TOP-$k$ SAMPLING

We first investigate the potential for expediting the `SpMM` operations in both the forward (Equation 2a) and backward (Equation 2b) passes with Top-$k$ sampling. More specifically, we substitute the forward and backward `SpMM` with their approximated counterparts in Equation 3. In this experiment, we set the $k$ as $0.1|\mathcal{V}|$ across different layers. We detail the model configuration in Appendix A.

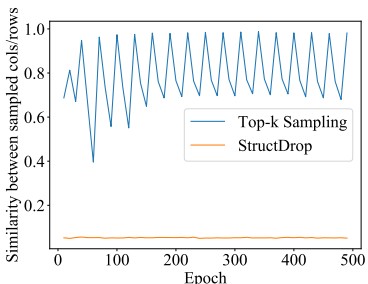

The performance results are presented in Table 1. As indicated by the results, we observed a substantial decrease in accuracy. This outcome is both surprising and intriguing, considering that theory Drineas et al. (2006) has previously demonstrated that Top-$k$ sampling should yield a satisfactory approximation with minimal reconstruction error to the original matrix multiplication. To dig in further, we examine the Jaccard similarity for the selected column/row pairs. We conduct this analysis using GCN training with the ogbn-Arxiv dataset as an example, and present the results in Figure 2. Upon closer inspection, we discovered that the Top-$k$ sampling consistently selects nearly identical column-row pairs in adjacent iterations. Specifically, the Jaccard similarity be-

Figure 2: The Jaccard Similarity of selected column-row pairs across the iterations in Top-$k$ Sampling. Top-$k$ incurs greatly repetative col/row pairs causing underfitting problem.

tween iterations in close proximity is approximately $90\%$. This suggests that the Top-$k$ sampling consistently utilizes the same subset of nodes and edges throughout graph learning. Consequently, a substantial portion of the graph information will be excluded during message aggregation, which leads to under-fitting problem.

To validate our hypothesis, we plot the training and test accuracy of a three-layer GCN model on ogbn-Products using various training schemes, as shown in Figure 3. The under-fitting hypothesis finds support in Figure 3a, where the training accuracy using Top-$k$ sampling is significantly lower compared to the baseline. As a consequence, Figure 3b shows that the test accuracy of GNNs trained with Top-$k$ sampling is also substantially inferior to the baseline.

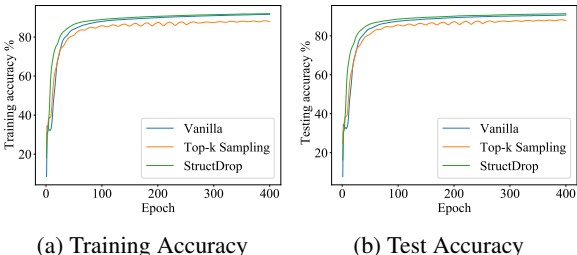

(a) Training Accuracy            (b) Test Accuracy

Figure 3: Training and testing accuracy comparison between different baselines on GCN with ogbn-Product.

### 3.2 STRUCTDROP: AN EFFICIENT SAMPLING SCHEME WITH INCREASED GENERALIZABILITY

Motivated by the observation that Top-$k$ sampling leads to under-fitting due to the consistent selection of the same graph information during training, we explore a straightforward strategy: uniform selection of each column-row pair. In other words, **each column-row pair has an equal probability of being sampled, and we sample a total of $k$ column-row pairs without replacement.** We call this simple yet effective strategy

Table 1: Preliminary results on three datasets. "+Top-$k$ Sampling" means we replace both the forward and backward `SpMM` with their approximated version. Here we set the $k$ as $0.1|\mathcal{V}|$ across different layers. All reported results are averaged over six random trials.

|  |  | Reddit | ogbn-Arxiv | ogbn-Product |
|---|---|---|---|---|
| GCN | Baseline | $95.30 \pm 0.05$ | $72.09 \pm 0.26$ | $76.05 \pm 0.10$ |
|  | +Top-$k$ Sampling | $93.53 \pm 0.44$ | $70.33 \pm 0.86$ | $74.73 \pm 1.81$ |
| GraphSAGE | Baseline | $96.59 \pm 0.03$ | $70.44 \pm 0.31$ | $78.05 \pm 0.90$ |
|  | +Top-$k$ Sampling | $90.35 \pm 1.22$ | $62.10 \pm 0.52$ | $70.17 \pm 0.32$ |

`StructDrop`, structurally sampling the whole graph. Experiments result in section 4.3 show that this structured sampling method yields better performance compared to the unstructured dropout approach. Here we analyze the potential of our method from a generalizability and efficiency perspective.

**Generalizability Analysis**   As demonstrated in Figure 2, `StructDrop` employs a varied set of column-row pairs throughout the training process, indicating that `StructDrop` effectively integrates information from the entire graph. From a different perspective, `StructDrop` eliminates entire columns in the adjacency matrix while leaving rows unchanged. This results in the removal of all outgoing edges for a specific set of nodes. The operation applied to such a sampled adjacency matrix and node embeddings introduces randomness during aggregation, which can be regarded as a form of data augmentation. Consequently, there is increased randomness and variability in the aggregated nodes, which enhances generalizability. As a result, both Figure 3a and Figure 3b illustrate that the training and test accuracy of `StructDrop` closely match those of the baseline. This suggests that `StructDrop` effectively mitigates the under-fitting issue.

**Efficiency Analysis**   Previous approaches have utilized edge/node dropping as data augmentation techniques to enhance generalizability. Such methods also appear to increase computing speed due to the FLOPs reduction, which is achieved by dropping entries in the adjacency matrix. However, these methods encounter efficiency challenges because the speedup gained from reducing FLOPs is often offset by the complex operations involved in manipulating the adjacency matrix.

Digging deeper, a graph can usually be represented by two data structures: the sparse adjacency matrix and edge index. The adjacency matrix can be viewed as a data structure optimized for computation time, and employing the adjacency matrix often leads to much faster computations compared to using the edge index format spm; pyg (2023). Nonetheless, a gap emerges because such computation-friendly data structure is usually represented in the Compress Sparse Row (CSR) format Arai et al. (2016), which cannot be easily manipulated due to the compression of the row indices. On the contrary, the edge index is an manipulation-friendly data structure that can be easily modified. Thus, edge/node dropping operations are typically carried out on the edge index dro (a;b). However, this process introduces time overhead because the data structure must be converted back to the computation-friendly adjacency matrix for faster computation. This additional conversion offsets the speed gains achieved through reduced FLOPs.

With the structured dropping approach, we can directly manipulate the computation-friendly adjacency matrix since we only drop the column-wise outgoing edges, which can be directly implemented upon the CSR format. Consequently, our method bypasses the conversion from edge indices to sparse adjacency matrix, resulting in fast sampling implementation. Our extensive experiment results in Sec 4.2 demonstrates that our structured dropping method achieves a substantial increase in efficiency when compared to the edge/node-oriented dropping methods. Importantly, this efficiency boost introduced in our method is achieved without sacrificing accuracy during training.

### 3.3 INSTANCE NORMALIZATION MEETS THE SAMPLING SCHEME

While the fast matrix multiplication with random sampling brings notable efficiency benefits, a side effect is the distribution shift of node embeddings during training. This shift arises due to the random sampling of column-row pairs between epochs, leading to the entirely different node embeddings learned from the diverse sets of neighbors. It is widely observed that such a sharp distribution shift can impede the learning rate and even steer the model towards the convergence of suboptimal points. Bjorck et al. (2018); Ioffe & Szegedy (2015); Bjorck et al. (2018).

To mitigate the training shift which causes the unstable convergence, we apply instance normalization at critical point following the approximated matrix multiplication. Mathematically, recalling the forward pass in Equation 2a, we use $\boldsymbol{H}^{(l)} = \texttt{SpMM}(\texttt{StructDrop}(\tilde{\boldsymbol{A}}, \boldsymbol{J}^{(l)}))$ to represent the node embeddings after neighbor aggregation. These embeddings are obtained by uniformly dropping the column-row pairs over matrices $\tilde{\boldsymbol{A}}$ and $\boldsymbol{J}^{(l)}$ and then performing sparse matrix production on them. Considering embedding vector $\boldsymbol{h}_i^{(l)} \in \mathbb{R}^d$ of node $v_i$, i.e., the $i^{\text{th}}$ row in $\boldsymbol{H}^{(l)}$, the instance normalization rescales it by Ulyanov et al. (2016):

$$\tilde{\boldsymbol{h}}_i^{(l)} = [\boldsymbol{h}_i^{(l)} - \text{E}(\boldsymbol{h}_i^{(l)})] \,/\, \text{Sqrt}(\text{Var}(\boldsymbol{h}_i^{(l)}) + \epsilon) * \boldsymbol{\gamma} + \boldsymbol{\beta}. \tag{5}$$

$\text{E}(\cdot)$, $\text{Sqrt}(\cdot)$, and $\text{Var}(\cdot)$ denote operations of expectation, squared root, and variance, respectively; $\boldsymbol{\gamma}, \boldsymbol{\beta} \in \mathbb{R}^d$ represents the trainable weights for the running variance and mean, respectively. Each node embedding is rescaled to mitigate the effects of sampling randomness, thereby facilitating the convergence of the model with improved generalization. Detailed experiments discussing node embedding shifting and generalization performance are provided in the experimental section 4.3 to substantiate our proposed approach.

## 4 EXPERIMENTS

In our experiments, we evaluate our proposed framework through answering the following research questions: **Q1:** How effectively is `StructDrop`'s generalizability? **Q2:** To what extent does `StructDrop` accelerate the training speed? **Q3:** How crucial is the role of instance normalization within the sampling scheme?

### 4.1 IMPLEMENTATION DETAILS

**Datasets, Backbones and Baselines**  To evaluate `StructDrop`, we adopt four large scale graph benchmarks which are commonly used in different domains: Reddit Hamilton et al. (2017a), Reddit2 Zeng et al. (2020) [1], ogbn-Arxiv Hu et al. (2020) and ogbn-Products Hu et al. (2020). We evaluate `StructDrop` using both the full-batch and sub-batch training settings. We intergate `StructDrop` with seven popular schemes in large graph training including GCN, GraphSAGE, GCNII, GIN and other subsampling based mechanism (GraphSAINT, GraphSAGE and ClusterGCN). The comparison are made against four different baselines introduced in Sec 4.2.2. We detail our hyperparameter settings in Appendix A.

### 4.2 SUPERIOR GENERALIZABILITY AND EFFICIENCY

In this section, we first evaluate the generalizability and efficiency of `StructDrop` in comparison to different baselines. As mentioned in Sec 3.3, `StructDrop` greatly accelerates the graph computation while simultaneously enhancing generalizability. This is evident from the negligible accuracy

---

[1]This is a sparser version of the original Reddit dataset ( 23M edges instead of  114M edges), and is used in paper GraphSAINT Zeng et al. (2020)

Table 2: Here we presents a comparison of efficiency and accuracy across different baseline methods using GCN, GraphSAGE, GIN, and sub-sampling based ClusterGCN. We observe that in most experiments, Top-k Sampling experiences a significant accuracy drop (over 1%, and in most cases exceeding 3%), which is highlighted in red. These accuracy reductions make it unsuitable for real-world deployment. For the speedup comparison, we exclude results where the accuracy drop is too severe (marked in red) and highlight the **best** speedup gains in bold. We note that `StructDrop` achieves the best speedup gain without accuracy loss compared to the other baselines. We provide additional results for GCNII and other subgraph sampling methods including GraphSAINT and GraphSAGE in Table 10 located in Appendix B.1.

| | # nodes | 232,965 | | 232,965 | | 169,343 | | 2,449,029 | |
| | # edges | 114,615,892 | | 23,213,838 | | 1,166,243 | | 61,859,140 | |
| Model | Methods | Reddit | | Reddit2 | | ogbn-Arxiv | | ogbn-Products | |
| | | Accuracy | Speedup | Accuracy | Speedup | Accuracy | Speedup | Accuracy | Speedup |
|---|---|---|---|---|---|---|---|---|---|
| GCN | Vanilla | $95.3 \pm 0.05$ | $1\times$ | $95.38 \pm 0.06$ | $1\times$ | $72.09 \pm 0.26$ | $1\times$ | $76.05 \pm 0.10$ | $1\times$ |
| | Top-k Sampling | $93.21 \pm 0.15$ | $6.99\times$ | $94.21 \pm 0.25$ | $2.72\times$ | $70.84 \pm 0.63$ | $1.33\times$ | $77.94 \pm 2.47$ | $\mathbf{1.96\times}$ |
| | DropEdge | $95.44 \pm 0.01$ | $1.87\times$ | $95.47 \pm 0.02$ | $1.72\times$ | $72.55 \pm 0.33$ | $1.21\times$ | $78.96 \pm 0.60$ | $1.2\times$ |
| | DropNode | $95.34 \pm 0.06$ | $2.07\times$ | $95.35 \pm 0.05$ | $1.7\times$ | $72.36 \pm 0.20$ | $1.23\times$ | $78.29 \pm 2.15$ | $1.17\times$ |
| | StructDrop | $95.47 \pm 0.05$ | $\mathbf{3.87\times}$ | $95.46 \pm 0.03$ | $\mathbf{2.4\times}$ | $72.46 \pm 0.23$ | $\mathbf{1.29\times}$ | $79.24 \pm 0.74$ | $1.8\times$ |
| GraphSAGE | Vanilla | $96.59 \pm 0.03$ | $1\times$ | $96.67 \pm 0.03$ | $1\times$ | $70.44 \pm 0.31$ | $1\times$ | $78.05 \pm 0.90$ | $1\times$ |
| | Top-k Sampling | $92.73 \pm 0.33$ | $9.66\times$ | $93.84 \pm 0.28$ | $3.08\times$ | $63.75 \pm 0.42$ | $1.39\times$ | $73.22 \pm 0.23$ | $3.31\times$ |
| | DropEdge | $96.65 \pm 0.03$ | $2.65\times$ | $96.55 \pm 0.03$ | $1.54\times$ | $70.23 \pm 0.19$ | $0.81\times$ | $78.57 \pm 0.09$ | $1.33\times$ |
| | DropNode | $96.36 \pm 0.06$ | $2.72\times$ | $96.33 \pm 0.01$ | $1.78\times$ | $69.99 \pm 0.29$ | $1.02\times$ | $78.93 \pm 0.20$ | $1.32\times$ |
| | StructDrop | $96.65 \pm 0.04$ | $\mathbf{4.26\times}$ | $96.56 \pm 0.03$ | $\mathbf{2.33\times}$ | $70.03 \pm 0.26$ | $\mathbf{1.15\times}$ | $78.97 \pm 0.17$ | $\mathbf{2.47\times}$ |
| GIN | Vanilla | $94.39 \pm 0.08$ | $1\times$ | $94.76 \pm 0.03$ | $1\times$ | $70.86 \pm 0.18$ | $1\times$ | $78.02 \pm 0.15$ | $1\times$ |
| | Top-k Sampling | $91.21 \pm 0.22$ | $2.45\times$ | $91.77 \pm 0.34$ | $2.33\times$ | $70.82 \pm 0.10$ | $1.16\times$ | $75.59 \pm 0.08$ | $1.34\times$ |
| | DropEdge | $94.54 \pm 0.07$ | $2.94\times$ | $94.83 \pm 0.08$ | $2.31\times$ | $71.11 \pm 0.15$ | $1.18\times$ | $78.65 \pm 0.13$ | $1.18\times$ |
| | DropNode | $94.41 \pm 0.05$ | $3.73\times$ | $94.69 \pm 0.01$ | $2.59\times$ | $70.64 \pm 0.12$ | $1.23\times$ | $78.16 \pm 0.19$ | $1.16\times$ |
| | StructDrop | $94.48 \pm 0.07$ | $\mathbf{5.29\times}$ | $94.86 \pm 0.03$ | $\mathbf{3.06\times}$ | $70.64 \pm 0.10$ | $\mathbf{1.28\times}$ | $78.73 \pm 0.05$ | $\mathbf{2.12\times}$ |
| ClusterGCN | Vanilla | $95.77 \pm 0.16$ | $1\times$ | $95.85 \pm 0.14$ | $1\times$ | $71.12 \pm 0.09$ | $1\times$ | $78.88 \pm 0.12$ | $1\times$ |
| | Top-k Sampling | $89.14 \pm 1.21$ | $1.61\times$ | $90.59 \pm 1.03$ | $1.25\times$ | $65.48 \pm 0.35$ | $1.16\times$ | $69.64 \pm 0.13$ | $1.17\times$ |
| | DropEdge | $95.73 \pm 0.09$ | $0.53\times$ | $95.62 \pm 0.11$ | $0.74\times$ | $71.07 \pm 0.36$ | $0.51\times$ | $78.72 \pm 0.02$ | $0.41\times$ |
| | DropNode | $95.71 \pm 0.05$ | $0.56\times$ | $95.72 \pm 0.07$ | $0.76\times$ | $70.62 \pm 0.19$ | $0.63\times$ | $76.36 \pm 0.43$ | $0.42\times$ |
| | StructDrop | $95.69 \pm 0.14$ | $\mathbf{1.36\times}$ | $95.60 \pm 0.05$ | $\mathbf{1.2\times}$ | $71.04 \pm 0.44$ | $\mathbf{1.12\times}$ | $78.34 \pm 0.03$ | $\mathbf{1.1\times}$ |

loss observed, coupled with significantly faster training speeds, as illustrated in our experimental results. We provide a detailed experimental findings below.

### 4.2.1 OPERATIONAL LEVEL ACCELERATION

We first evaluate the speed improvements at the operation level introduced by `StructDrop`. Figure 1 illustrates the speed improvements at the operation level achieved by `StructDrop`. We measured the wall clock completion time of various operators across different datasets. With `StructDrop`, the computational complexity in sparse matrix multiplication is significantly reduced in a hardware-friendly way, resulting in faster completion times. Across datasets, the forward pass `SpMM` operation is accelerated by 1.9 to 5.5 times, while the backward pass `SpMM` is accelerated by a factor of 2.62 to 4.8 times. Overall, `StructDrop` achieves a maximum wall clock time speedup of $5.29\times$ compared to the vanilla baseline as shown in table 2.

### 4.2.2 END-TO-END PERFORMANCE ANALYSIS

Next, we assess the end-to-end training speedup and model accuracy of `StructDrop` in comparison to different methods. Specifically, we compare our approach against: *1,* Vanilla baseline with the standard training process without any approximations; *2,* Top-$k$ sampling Adelman et al. (2021) and *3,* DropEdge Rong et al. (2019) and DropNode Feng et al. (2020). We conduct the experiments with the same sampling ratio across all different baselines to ensure a fair comparison. We present the results on GCN, GraphSAGE, GIN and subgraph sampling based ClusterGCN in Table 2. Due to space limitation, we put additional results regarding GCNII and other subgraph sampling based method (GraphSAINT, GraphSAGE) in Table 10 in Appendix B.1 for further details.

***StructDrop* achieves much faster speed with almost no accuracy drop or even better accuracy** `StructDrop` achieves remarkable speedup with negligible accuracy loss (within 0.5%) or even better accuracy compared to vanilla training scheme. As discussed in Sec 3.2, the maintained or enhanced accuracy is attributed to `StructDrop`'s random sampling during the message aggregation phase. These samples introduce randomness, effectively acting as data augmentation, which enhances `StructDrop`'s generalizability. We defer more discussion in generalizability in Sec 4.2.3.

In terms of efficiency, `StructDrop` achieves an end-to-end wall clock training completion time speedup of up to 5.29 times compared to the vanilla baseline as shown in Table 2. This speedup is derived from the fast approximation operation during message aggregation, which significantly reduces computational complexity without introducing additional overhead. In summary, `StructDrop` represents a novel and effective acceleration scheme that enhances the efficiency of GNN training while preserving accuracy. We now compare our training scheme with other baselines.

***Notable accuracy improvement compared to Top-$k$ sampling:*** We now compare `StructDrop` with Top-$k$ sampling. We highlight the significant accuracy improvement achieved by `StructDrop` here. As shown in table 2, Top-$k$ sampling results in an unacceptable performance loss compared to both the vanilla baseline and `StructDrop`. This performance degradation is attributed to Euclidean norm-based sampling, which tends to overly concentrate on a few columns and rows, as evident in our profiled Jaccard similarity analysis

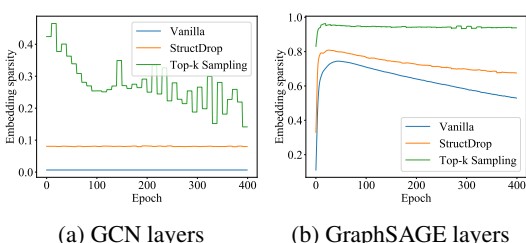

(a) GCN layers    (b) GraphSAGE layers

Figure 4: Embedding sparsity during training

shown in Figure 2. Consequently, this leads to the loss of global graph information during message aggregation and contributes to the underfitting behavior.

In contrast, the uniform random sampling strategy employed in `StructDrop` results in the collection and utilization of global graph knowledge during message aggregation, as every column-row pair has the potential to be involved. This approach facilitates more comprehensive graph learning.

Another significant factor to the poor performance of Top-$k$ sampling is the information loss that occurs during training. We conducted profiling of the embedding sparsity after message aggregation with vanilla, Top-$k$ and `StructDrop` shown in Figure 4. We found that after sampling and message passing, the embeddings obtained through the Top-$k$ sampling exhibit a high rate of zero entries. Although Euclidean norm-based sampling maintains minimal reconstruction error when compared to vanilla sparse matrix multiplication, it tends to select cols/rows with lower degrees Liu et al. (2023a). This selection results in higher sparsity and consequently leads to more significant information loss during aggregation, exacerbating the underfitting problem.

As depicted in Figure 4, the embedding sparsity of `StructDrop` is comparable to that of the vanilla scheme, resulting in less information loss during message passing. In Appendix C, we further demonstrate that under the same accuracy requirements, `StructDrop` achieves better accuracy and speedup compared to Top-$k$ sampling. In summary, `StructDrop` outperforms the Top-$k$ sampling scheme with significantly better accuracy.

***Considerably faster training speed compared to DropEdge and DropNode:*** DropEdge Rong et al. (2019) is a method designed to address overfitting and oversmoothing issues in GNN training. On the other hand, DropNode Feng et al. (2020) utilizes node feature random dropouts as a form of data augmentation to enhance robust training. DropEdge and DropNode randomly sample edges or nodes in the input graph based on certain probabilities. As indicated in Table 2, `StructDrop` achieves comparable accuracy (within 0.5%) to both DropEdge and DropNode across different datasets. This highlights the effectiveness of data augmentation through sampled message passing.

However, `StructDrop`'s true strength lies in its substantial efficiency gains compared to the other two baselines. Table 3 shows the speedup gain of `StructDrop` on GraphSAGE. Overall `StructDrop` can achieve up to 2.07x and 2.42x speedup compared to DropEdge and DropNode re-

Table 3: `StructDrop`'s speedup benefit vs. DropEdge and DropNode

|  | Reddit | Reddit2 | ogbn-Arxiv | ogbn-Products |
|---|---|---|---|---|
| vs. DropEdge | 1.61 × | 1.51 × | 1.42 × | 1.86 × |
| vs. DropNode | 1.57 × | 1.31 × | 1.13 × | 1.87 × |

spectively, primarily driven by hardware efficiency. While the number of preserved edges during training remains consistent, DropEdge and DropNode exhibit significantly smaller dropping granularity compared to `StructDrop`. Manipulating such sampling operations incurs additional conversion overhead, as discussed in Sec 3.2. In contrast, `StructDrop`'s random dropping operation on all the outgoing edges in the entire columns can be applied directly to the computation-friendly adjacency matrix. This faster sampling introduces almost no additional performance overhead while expediting graph training with much faster computation, ultimately translating into speed improvements.

***StructDrop** acceleration effect on full-graph and subgraph training.* `StructDrop` is a mechanism for column and row sampling during graph training, which can be seamlessly integrated into both full-graph and subgraph-based training. We observe that `StructDrop` achieves more significant speedup in full-graph training. Additionally, the speedup effect scales as the size of the subgraph increases. More details from our ablation study can be found in Table 11 in Sec B.2. In real-world scenarios, subgraphs are typically large to retain more global information and improve hardware efficiency. Nevertheless, `StructDrop` can substantially accelerate graph training for both full-graph and subgraph-based approaches.

In general, `StructDrop` achieves superior speedup (up to 5.29x) with negligible drop or even more exciting results on accuracy, as shown in Table 2 and 10. While the ratio of speedup varies, the speedup effect remains consistent across all different architectures and datasets, and we provide a detailed discussion of these variations in speedup gain in Appendix B.3.

### 4.2.3 GENERALIABILITY STUDY OF STRUCTDROP

In this section, we aim to gain a deeper understanding of `StructDrop`'s generalizability. We begin by using ogbn-Products as an example to plot the training loss and generalization gap for different baselines and GNN architectures in Figure 5 and 7. The generalization gap is quantified as the difference between the training and testing loss, with a higher loss gap indicating better generalizability. Despite the Top-$k$ sampling mechanism exhibiting the highest training loss and underfitting during training with the GCN, `StructDrop` achieves the largest generalization gap. These results are consistent with previous analysis, suggesting that randomness and diversity introduced by `StructDrop` act as a form of data augmentation, thereby enhancing the model's generalizability.

### 4.2.4 ABLATION STUDIES OF DROPPING RATIO

In this section, we provide a comprehensive analysis of `StructDrop` with respect to the dropping ratio using GCN as an example. We also included the results of other backbones in Appendix D.

Table 4 presents `StructDrop`'s performance across different sampling ratios and datasets on GCN. The impact of the sample ratio on accuracy varies depending on the datasets. For smaller datasets like ogbn-Arxiv which contain a small number of edges, higher sample ratios tend to lead to higher accuracy, as there is less information loss. Conversely, for larger datasets like ogbn-Products which potentially have more information redundancy due to the large number of edges, accuracy is inversely proportional to the sample ratio. This is because redundant edges can cause the node embeddings to be smoothed by their neighbors, resulting in a loss of node features with the converged embeddings. Regarding efficiency, lower sampling ratios result in higher computation speeds. The trends for GraphSAGE and other model architectures are similar.

Table 4: Accuracy and speedup on different sample ratios

| Model | Ratio | Reddit | | Reddit2 | | ogbn-Arxiv | | ogbn-Products | |
|---|---|---|---|---|---|---|---|---|---|
| | | Accuracy | Speedup | Accuracy | Speedup | Accuracy | Speedup | Accuracy | Speedup |
| GCN | 0.1 | $95.44 \pm 0.04$ | $5.63 \times$ | $95.39 \pm 0.05$ | $2.81 \times$ | $72.16 \pm 0.21$ | $1.35 \times$ | $79.51 \pm 1.07$ | $2.04 \times$ |
| | 0.2 | $95.47 \pm 0.05$ | $3.87 \times$ | $95.46 \pm 0.03$ | $2.40 \times$ | $72.46 \pm 0.23$ | $1.29 \times$ | $79.24 \pm 0.74$ | $1.8 \times$ |
| | 0.3 | $95.47 \pm 0.04$ | $2.89 \times$ | $95.48 \pm 0.03$ | $2.05 \times$ | $72.44 \pm 0.24$ | $1.22 \times$ | $78.95 \pm 0.46$ | $1.6 \times$ |
| | 0.4 | $95.43 \pm 0.04$ | $2.26 \times$ | $95.46 \pm 0.04$ | $1.78 \times$ | $72.66 \pm 0.23$ | $1.17 \times$ | $78.63 \pm 0.29$ | $1.43 \times$ |

### 4.3 BENEFITS OF INSTANCE NORMALIZATION IN SAMPLING

We further evaluate the advantages with incorporating instance normalization during sampling. Instance norm serves as a mitigator of distribution shifts, reducing the shifts in embeddings induced by random sampling between epochs. The results presented in Figure 6 demonstrate that instance norm serves as an effective factor in smoothing the training process, ultimately leading to improved accuracy.

***Ablation Study of Instance Norm*** We evaluate the accuracy improvement resulting from the inclusion of instance norm. We summarize the accuracy using GCN and GraphSAGE as examples on different datasets w/o instance norm applied. As depicted in Table 5, the accuracy with instance

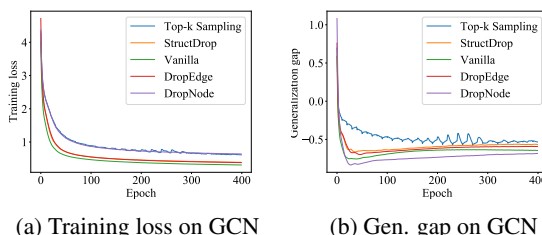

(a) Training loss on GCN     (b) Gen. gap on GCN

Figure 5: Training curve on ogbn-Products dataset

Table 5: Ablation study of instance normalization.

|  |  | Reddit | ogbn-Arxiv | ogbn-Products |
|---|---|---|---|---|
| GCN | w/ instance norm | $95.47 \pm 0.05$ | $72.46 \pm 0.23$ | $79.24 \pm 0.74$ |
|  | w/o instance norm | $94.01 \pm 1.04$ | $69.30 \pm 1.19$ | $74.55 \pm 3.51$ |
| GraphSAGE | w/ instance norm | $96.65 \pm 0.04$ | $70.03 \pm 0.26$ | $78.97 \pm 0.17$ |
|  | w/o instance norm | $96.52 \pm 0.04$ | $69.00 \pm 0.45$ | $78.25 \pm 0.21$ |

norm applied is consistently higher than that without it across datasets. Instance norm is beneficial for random sampling, resulting in improved accuracy.

***Effect for Smooth Training*** Next we deep dive into why instance norm helps boost the accuracy. We plot the distribution shift of the embedding after message aggregation with sampled columns/rows in Figure 6. We use the norm difference of the embedding between subsequent epochs to measure the training smoothness. As shown in Figure 6, training without instance norm causes much larger embedding shifts, making the training process not smooth as the model needs to con-

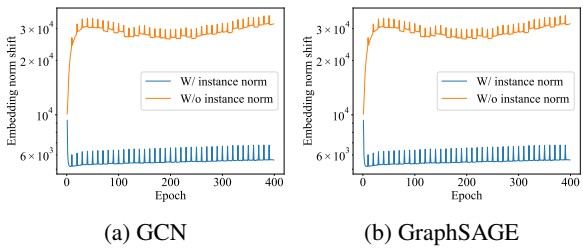

(a) GCN     (b) GraphSAGE

Figure 6: Embedding shifts between epochs

stantly adapt to new inputs distribution. This effect exacerbates as the random samples causes message aggregation in different epochs varies drastically. Instance norm successfully lowers the embedding shifts, thus stabilize the training process and leads to better accuracy.

## 5 RELATED WORK

**Large-scale Graph Learning**   Massage passing over graph can described by sparse matrix multiplication. Such operation is resource consuming, where the memory and time complexities depend on the amounts of nodes and edges, respectively. To address the scalability issue, numerous families of algorithms have been explored, including the subgraph-based GNN training Hamilton et al. (2017a); Huang et al. (2018) , graph precomputation Wu et al. (2019); Klicpera et al. (2018); Yu et al. (2020), and distributed training Zha et al. (2023; 2022); Yuan et al. (2022); Wang et al. (2022). The common merit of them is to divide the large graph into pieces, each of which could be handled by the resource-limited GPU.

Related work on **Efficient Training Algorithms**, **Subgraph Sampling**, **Random Dropout**, **Graph Condensation** and other topics are also important. Due to space limitations, we defer the discussion on them to Appendix F.

## 6 CONCLUSIONS

In our work, we introduce `StructDrop` to replace time-consuming message passing with fast sparse matrix multiplication (`SpMM`) during whole training process of GNNs. `StructDrop` uniformly samples column-row pairs in the adjacency matrix, reducing computational complexity in `SpMM`. To address distribution shifts resulting from random sampling, we apply instance norm after `SpMM` to rescale node embeddings and stabilize the training. Extensive experiments on benchmarks confirm the effectiveness of our approach that achieves a superior performance on efficiency and generalization.

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

# A  CONFIGURATION AND HYPERPARAMETER SETTING

`StructDrop` only has one hyperparameter which is the sampling ratio. We present comprehensive sample ratio ablation study in Sec 4.2.4. We adopt a similar approach to prior study Liu et al. (2023a) by sampling every ten training steps. Below tables show the configurations of different model architectures (GCN, GraphSAGE, GCNII and GraphSAINT) in graph training.

Table 6: Configuration of Full-Batch GCN.

| Dataset | Training | | | Archtecture | | |
|---|---|---|---|---|---|---|
| | Learning Rates | Epochs | Dropout | BatchNorm | Layers | Hidden Dimension |
| Reddit | 0.01 | 400 | 0.5 | No | 3 | 256 |
| Reddit2 | 0.01 | 400 | 0.5 | No | 3 | 256 |
| ogbn-Arxiv | 0.01 | 500 | 0.1 | No | 3 | 512 |
| ogbn-Products | 0.001 | 400 | 0.5 | No | 3 | 256 |

Table 7: Configuration of Full-Batch GraphSAGE.

| Dataset | Training | | | Archtecture | | |
|---|---|---|---|---|---|---|
| | Learning Rates | Epochs | Dropout | BatchNorm | Layers | Hidden Dimension |
| Reddit | 0.01 | 400 | 0.5 | No | 3 | 256 |
| Reddit2 | 0.01 | 400 | 0.5 | No | 3 | 256 |
| ogbn-Arxiv | 0.01 | 500 | 0.1 | No | 3 | 512 |
| ogbn-Products | 0.001 | 500 | 0.5 | No | 3 | 256 |

Table 8: Configuration of Full-Batch GCNII.

| Dataset | Training | | | Archtecture | | |
|---|---|---|---|---|---|---|
| | Learning Rates | Epochs | Dropout | Alpha&Theta | Layers | Hidden Dimension |
| Reddit | 0.01 | 400 | 0.5 | 0.1&0.5 | 4 | 256 |
| Reddit2 | 0.01 | 400 | 0.5 | 0.1&0.5 | 4 | 256 |
| ogbn-Arxiv | 0.01 | 500 | 0.1 | 0.1&0.5 | 4 | 512 |
| ogbn-Products | 0.001 | 500 | 0.1 | 0.1&0.5 | 3 | 128 |

Table 9: Configuration of GraphSAINT.

| Dataset | Training | | | Archtecture | | |
|---|---|---|---|---|---|---|
| | Learning Rates | Epochs | Dropout | Walk length | Layers | Hidden Dimension |
| Reddit | 0.01 | 40 | 0.1 | 4 | 3 | 512 |
| Reddit2 | 0.01 | 40 | 0.1 | 4 | 3 | 512 |
| ogbn-Arxiv | 0.01 | 75 | 0.1 | 4 | 4 | 512 |
| ogbn-Products | 0.01 | 20 | 0.5 | 3 | 3 | 256 |

Table 10: We presents additional experiment results for comparison of efficiency and accuracy across different baseline methods with GCNII and subsampling methods like GraphSAINT and GraphSAGE. Consistent with the phenominon we observed in table 2, Top-k Sampling experiences a significant accuracy drop (over 1%, and in most cases exceeding 3%), which is highlighted in red. These accuracy reductions make it unsuitable for real-world deployment. For the speedup comparison, we exclude results where the accuracy drop is too severe (marked in red) and highlight the **best** speedup gains in bold. Consistently, StructDrop achieves the best speedup gain without accuracy loss compared to the other baselines.

| # nodes | | 232,965 | | 232,965 | | 169,343 | | 2,449,029 | |
|---|---|---|---|---|---|---|---|---|---|
| # edges | | 114,615,892 | | 23,213,838 | | 1,166,243 | | 61,859,140 | |
| Model | Methods | Reddit | | Reddit2 | | ogbn-Arxiv | | ogbn-Products | |
| | | Accuracy | Speedup | Accuracy | Speedup | Accuracy | Speedup | Accuracy | Speedup |
| GCNII | Vanilla | $96.81 \pm 0.03$ | $1 \times$ | $96.80 \pm 0.02$ | $1 \times$ | $72.12 \pm 0.24$ | $1 \times$ | $76.70 \pm 0.12$ | $1 \times$ |
| | Top-k Sampling | $91.46 \pm 1.00$ | $5.14 \times$ | $93.51 \pm 0.58$ | $2.11 \times$ | $71.09 \pm 0.09$ | $1.21 \times$ | $74.27 \pm 0.34$ | $1.74 \times$ |
| | DropEdge | $96.81 \pm 0.07$ | $2.02 \times$ | $96.72 \pm 0.01$ | $1.61 \times$ | $72.24 \pm 0.30$ | $1.14 \times$ | $77.49 \pm 0.09$ | $1.02 \times$ |
| | DropNode | $96.39 \pm 0.05$ | $2.16 \times$ | $96.31 \pm 0.03$ | $1.63 \times$ | $72.35 \pm 0.01$ | $1.13 \times$ | $77.72 \pm 0.18$ | $1.01 \times$ |
| | StructDrop | $96.82 \pm 0.02$ | $\mathbf{3.43} \times$ | $96.72 \pm 0.03$ | $\mathbf{1.97} \times$ | $72.16 \pm 0.12$ | $\mathbf{1.19} \times$ | $77.55 \pm 0.31$ | $\mathbf{1.62} \times$ |
| GraphSAINT | Vanilla | $95.85 \pm 0.13$ | $1 \times$ | $96.22 \pm 0.05$ | $1 \times$ | $70.72 \pm 0.17$ | $1 \times$ | $78.67 \pm 0.23$ | $1 \times$ |
| | Top-k Sampling | $90.36 \pm 0.84$ | $1.56 \times$ | $91.27 \pm 0.50$ | $1.08 \times$ | $65.77 \pm 0.41$ | $1.11 \times$ | $75.59 \pm 0.37$ | $1.33 \times$ |
| | DropEdge | $95.92 \pm 0.06$ | $0.7 \times$ | $96.12 \pm 0.03$ | $0.67 \times$ | $69.56 \pm 0.06$ | $0.79 \times$ | $79.50 \pm 0.18$ | $0.53 \times$ |
| | DropNode | $95.73 \pm 0.08$ | $0.73 \times$ | $96.05 \pm 0.11$ | $0.68 \times$ | $69.47 \pm 1.08$ | $0.82 \times$ | $79.27 \pm 0.33$ | $0.52 \times$ |
| | StructDrop | $95.87 \pm 0.05$ | $\mathbf{1.33} \times$ | $96.09 \pm 0.03$ | $\mathbf{1.05} \times$ | $69.40 \pm 0.94$ | $\mathbf{1.07} \times$ | $79.59 \pm 0.37$ | $\mathbf{1.27} \times$ |
| GraphSAGE | Vanilla | $96.47 \pm 0.10$ | $1 \times$ | $96.53 \pm 0.04$ | $1 \times$ | $70.49 \pm 0.29$ | $1 \times$ | $78.67 \pm 0.16$ | $1 \times$ |
| | Top-k Sampling | $93.19 \pm 1.42$ | $1.23 \times$ | $94.04 \pm 0.10$ | $1.26 \times$ | $62.85 \pm 2.34$ | $1.11 \times$ | $76.47 \pm 0.34$ | $1.2 \times$ |
| | DropEdge | $94.57 \pm 0.13$ | $0.92 \times$ | $95.92 \pm 0.11$ | $0.89 \times$ | $68.57 \pm 0.18$ | $0.87 \times$ | $79.40 \pm 0.21$ | $0.49 \times$ |
| | DropNode | $95.12 \pm 0.15$ | $0.92 \times$ | $96.11 \pm 0.09$ | $0.92 \times$ | $69.34 \pm 0.61$ | $0.88 \times$ | $78.81 \pm 0.44$ | $0.52 \times$ |
| | StructDrop | $96.34 \pm 0.08$ | $\mathbf{1.28} \times$ | $96.49 \pm 0.02$ | $\mathbf{1.23} \times$ | $69.2 \pm 0.56$ | $\mathbf{1.12} \times$ | $78.90 \pm 0.17$ | $\mathbf{1.21} \times$ |

# B  EFFICIENCY AND ACCURACY COMPARISON BETWEEN BASELINES ON GCNII AND SUBSAMPLING MECHANISMS

## B.1  PERFORMANCE ANALYSIS ON EFFICIENCY AND ACCURACY

Here we presents additonal results regarding StructDrop's accuracy and efficiency. The comparison between our StructDrop with other backbones with GCNII and subsampling mechanism (GraphSAINT, GraphSAGE) is shown in Table 10. The results shown in the table are consistent with the discussion in Sec 4.2.2. Take GCNII result as an example, StructDrop achieves a 3.43x speedup without compromising accuracy compared to the vanilla training scheme. Moreover, in subsampling-based experiments, our method achieves a 1.33x speedup in GraphSAINT AND a 1.28x speedup with GraphSAGE. The Top-$k$ method experiences a significant accuracy drop compared to all baselines. Additionally, StructDrop surpasses both DropEdge and DropNode methods in terms of speedup due to its computation-friendly dropping approach. These findings are consistent with other experiments presented in the main paper, elaborated in Sec 4.2.2.

## B.2  DETAILED ANALYSIS OF STRUCTDROP'S PERFORMANCE IN SUBGRAPH TRAINING

For the subgraph sampling scheme, we found the subgraph size affects the speedup gain. we conduct a further ablation study on input subgraph size and show the results in Table 11. The input subsampled graph size is proportional to some hyper-parameters such as random walk length and batch sizes in GraphSAINT. We use Reddit/Reddit2 dataset and train the model based on the GraphSAINT-

Table 11: Ablation study on StructDrop's acceleration effects with random walk length in GraphSAINT. Larger walk length will result in larger subgraph in GraphSAINT.

| | Walk length | 4 | 8 | 16 |
|---|---|---|---|---|
| Reddit | Speedup | 1.33x | 1.47x | 1.6x |
| | Accuracy | $95.87 \pm 0.05$ | $96.32 \pm 0.02$ | $95.97 \pm 0.08$ |
| Reddit2 | Speedup | 1.05x | 1.24x | 1.43x |
| | Accuracy | $96.09 \pm 0.03$ | $96.47 \pm 0.06$ | $96.20 \pm 0.02$ |

based method. We study the speedup gain with different random walk lengths. In this experiment, a larger random walk length leads to a larger subgraph, maintaining more global information during training. As shown in below table, we see that the speedup gain increased from 1.33 to 1.6 on Reddit, and respectfully 1.05 to 1.43 on Reddit2 when the walk length is larger. That being said, the `StructDrop` acceleration effect scales up when the subgraph is larger. Such speedup gain enabled by `StructDrop` is non-trivial. In the real-world setting, the size of the input subgraph is typically large. There are two considerations: 1. From GNN training perspective, a larger subgraph will preserve more global information, reducing information loss in the graph; 2. From the training efficiency side, it needs sufficient batches to keep the hardware fully occupied. With large graph, speeding up incurred in training will significantly save the training time and hardware resources, which could bring benefits and bring down the costs during training.

### B.3 SPEEDUP GAIN PERCENTAGE DIFFERENCE BETWEEN ARCHITECTURES AND DATASETS

As discussed in Sec 4.2.2, `StructDrop`'s consistently speedup the training among different architectures and datasets. There are percentage different in acceleration among datasets/architectures. We detail the explanation here. `StructDrop`'s operation-level acceleration (specifically, message passing operation acceleration as mentioned in Sec 4.2.1, which is an efficiency bottleneck during training) remains consistent across different architectures. However, different backbones might incur other operations other than the message passing (i.e. different linear layer dimensions). These operations are not accelerated and their overheads varies between backbones. Consequently, the percentage of acceleration differs across architectures. To further explain, if the operation-level acceleration is $p$, the overall speedup gain can be denoted as ($p$ * Overhead_OP + Overhead_Other) / (Overhead_OP + Overhead_Other), which will vary depending on different architectures. Similarly, different datasets with different size of the input graph will cause varying overhead. Nonetheless, `StructDrop` is able to speed the most inefficient message aggregation as mentioned in Sec 4.2.1, and the end to end speedup effect is consistent among different architectures and datasets as shown in Table 2 and 10.

## C DISCUSSION ON THE CHOICE OF TOP-$k$ AND STRUCTDROP UNDER RELAXED ACCURACY REQUIREMENTS.

As discussed in Sec 4.2.2, Top-$k$ method results in large accuracy drop ($\sim$8%) in some cases due to the under-fitting problem. Nevertheless, one might be curious how should Top-$k$ and `StructDrop` be chosen under a relaxed accuracy requirements ($\sim$2%). Under a loose accuracy requirements, although top-k method is in general faster (with lower accuracy), we would like to point out that the practitioner can accelerate `StructDrop` by reducing the percentage of columns/rows sampled in computation. We provide some experimental results as a comparison in the below Table 12. We use Reddit2 and Arxiv dataset with GCN dataset as the demonstration. Note that the Top-$k$'s accuracy is compromised a lot compared to Vanilla solution. We reduce the sample ratio of `StructDrop` in this experiment to check whether the speedup can catch up with the Top-$k$ mechanism.

Table 13: Ablation study on accuracy and speedup with different sample ratios on GraphSAGE, GCNII and GraphSAINT architecture

| Model | Ratio | Reddit | | Reddit2 | | ogbn-Arxiv | | ogbn-Products | |
|---|---|---|---|---|---|---|---|---|---|
| | | Acc. | Speedup | Acc. | Speedup | Acc. | Speedup | Acc. | Speedup |
| GraphSAGE | Vanilla | 96.59 ± 0.03 | 1 × | 96.67 ± 0.03 | 1 × | 70.44 ± 0.31 | 1 × | 78.05 ± 0.90 | 1 × |
| | 0.1 | 96.53 ± 0.04 | 6.48 × | 96.42 ± 0.04 | 2.93 × | 68.83 ± 0.30 | 1.33 × | 79.29 ± 0.07 | 2.96 × |
| | 0.2 | 96.65 ± 0.04 | 4.26 × | 96.56 ± 0.03 | 2.33 × | 70.03 ± 0.26 | 1.15 × | 78.97 ± 0.17 | 2.48 × |
| | 0.3 | 96.69 ± 0.04 | 3.13 × | 96.63 ± 0.04 | 2.01 × | 70.35 ± 0.24 | 1.12 × | 78.63 ± 0.12 | 2.1 × |
| | 0.4 | 96.68 ± 0.02 | 2.42 × | 96.67 ± 0.03 | 1.79 × | 70.65 ± 0.34 | 1.06 × | 78.31 ± 0.09 | 1.81 × |
| GCNII | Vanilla | 96.81 ± 0.03 | 1 × | 96.80 ± 0.02 | 1 × | 72.12 ± 0.24 | 1× | 76.70 ± 0.12 | 1 × |
| | 0.1 | 96.72 ± 0.03 | 4.61 × | 96.65 ± 0.03 | 2.19 × | 71.52 ± 0.07 | 1.24 × | 77.50 ± 0.35 | 1.77 × |
| | 0.2 | 96.82 ± 0.02 | 3.43 × | 96.72 ± 0.03 | 1.97 × | 72.16 ± 0.12 | 1.19 × | 77.55 ± 0.31 | 1.62 × |
| | 0.3 | 96.84 ± 0.03 | 2.67 × | 96.76 ± 0.03 | 1.77 × | 72.22 ± 0.21 | 1.15 × | 77.50 ± 0.31 | 1.49 × |
| | 0.4 | 96.85 ± 0.01 | 2.16 × | 96.80 ± 0.03 | 1.59 × | 72.20 ± 0.15 | 1.11 × | 77.25 ± 0.18 | 1.37 × |
| GraphSAINT | Vanilla | 95.85 ± 0.13 | 1 × | 96.22 ± 0.05 | 1 × | 70.72 ± 0.17 | 1 × | 78.67± 0.23 | 1 × |
| | 0.1 | 95.75 ± 0.08 | 1.47 × | 95.89 ± 0.01 | 1.1 × | 68.94 ± 0.62 | 1.13 × | 79.42 ± 0.12 | 1.34 × |
| | 0.2 | 95.87 ± 0.05 | 1.33 × | 96.09 ± 0.03 | 1.05 × | 69.40 ± 0.94 | 1.07 × | 79.59 ± 0.37 | 1.27 × |
| | 0.3 | 95.88 ± 0.03 | 1.23 × | 96.14 ± 0.05 | 1.03 × | 70.25 ± 0.92 | 1.05 × | 79.41 ± 0.31 | 1.18 × |
| | 0.4 | 96.01 ± 0.08 | 1.09 × | 96.19 ± 0.04 | 1.01 × | 70.49 ± 0.58 | 1.01 × | 79.21 ± 0.29 | 1.1 × |

From Table 12, we can see that by reducing the percentage of the columns/rows sampled during training, StructDrop's speedup gain can be effectively increased. With that, StructDrop successfully suppressed Top-$k$ at speed while still maintaining a much more superior accuracy. That's why a practitioner should choose StructDrop under a relaxed accuracy requirement.

Table 12: Comparison on efficiency and accuracy between Top-$k$ and StructDrop under relaxed accuracy requirements. **Bold** denotes the highest.

| | Method | Sample Ratio | Accuracy | Speedup compare to Vanilla |
|---|---|---|---|---|
| Reddit2 | Top-k | 0.1 | 94.21 ± 0.25 | 2.72 × |
| | StructDrop | 0.2 | **95.39 ± 0.05** | **2.81 ×** |
| ogbn-Arxiv | Top-k | 0.1 | 70.84 ± 0.63 | 1.33 × |
| | StructDrop | 0.2 | **72.16 ± 0.21** | **1.35 ×** |

At the same time, we believe the accuracy of the model is also important. StructDrop can effectively increase the training speed, with negligible accuracy loss or even more exciting accuracy in most cases. However, the model trained with top-k method suffers a lot (sometimes with ∼8%) for accuracy. Although faster, the experimental results (Table 2) show that Top-$k$ compromise the accuracy too much, which will cause large trouble during inference/model serving time. This is why we would like to advocate for training using StructDrop even with relaxed accuracy requirement.

## D   ABLATION STUDY ON ACCURACY AND EFFICIENCY WITH RATIO

The relationship between sampling ratios with respect to accuracy and efficiency of StructDrop is shown in Table 13. The results is consistent with the elaboration in Sec 4.2.4. The impact of the sample ratio on accuracy varies depending on the datasets. For smaller datasets, higher sample ratios tend to lead to higher accuracy because of less information loss. On the other hand, larger datasets like ogbn-Products which potentially have more information redundancy due to the large number of edges, accuracy could be inversely proportional to the sample ratio because those redundant edges can cause the node embeddings to be smoothed, which causes converged embeddings. For efficiency, lower sampling ratios result in higher computation speeds, and the trends for GraphSAGE and other model architectures are similar.

## E   GENERALIZATION ABILITY STUDY ON GRAPHSAGE

The training curve and generalization gap on GraphSAGE training on ogbn-Products dataset is shown in Figure 7. Similar to the result discussed in Sec 4.2.3, despite Top-$k$ with the highest training loss, StructDrop achieves the highest generalization gap owing to the randomness and

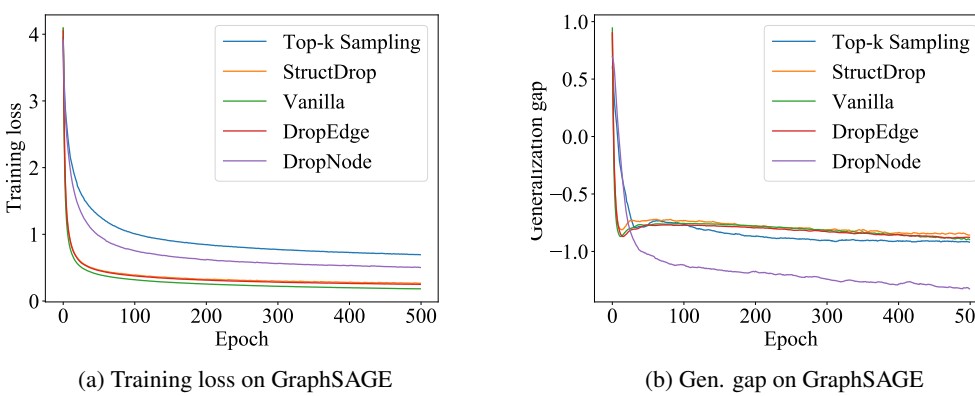

(a) Training loss on GraphSAGE

(b) Gen. gap on GraphSAGE

Figure 7: Training curve on GraphSAGE with ogbn-Products dataset.

diversity introduced by `StructDrop`, which act as a form of data augmentation, and thereby enhancing the model's generalizability.

## F MORE RELATED WORK

**Efficient Training Algorithms**    Another orthogonal line is to reduce the memory and time consumption by approximating the message passing. This can be divided into two categories. First, the adjacency matrix based approximation aims to compress the non-zero entries or matrix dimension. For example, Sketch-GNN sketch the graph adjacency matrix into a smaller one using hashing Chamberlain et al. (2022); DSpar expurgates the non-zero elements based on node degrees to obtain a sparse substitute Liu et al. (2023b). Second, the node embedding based approximation targets at compress the memory storage of hidden representations. For example, EXACT stocastically quantizes the node embeddings into low precision Liu et al. (2022); GNNAutoScale stores the whole list of node embeddings in CPU and retrieve them in forward propagation Fey et al. (2021).

**Random Dropout**    To improve the generalization performance on graph, there are two main categories of dropout. Edge-oriented dropout randomly samples a subset of edges to avoid over fitting and over-smoothing, such as DropEdge Rong et al. (2019), DropNode Feng et al. (2020), etc. On the other hand, Node-oriented dropout removes node features and links connected to the dropped nodes. The node-oriented dropout is originally motivated in sampling subgraph for scalable training and in augmenting graphs for contrastive learning, such as DropNode Feng et al. (2020), FastGCN Chen et al. (2018), etc.

**Subgraph-based GNN training**    This line of works focuses on training GNNs using sampled subgraphs to minimize the number of nodes stored in memory. Several sampling techniques have been developed based on this concept, such as node-wise sampling Hamilton et al. (2017a); Chen et al. (2017), layer-wise sampling Huang et al. (2018); Zou et al. (2019), and subgraph sampling Chiang et al. (2019); Zeng et al. (2019). `StructDrop` is a technique that performs row and column sampling on adjacency matrices during graph training, and it can be seamlessly combined with the previously mentioned subgraph sampling methods. Our experiments demonstrate that `StructDrop` improves computational efficiency while maintaining accuracy.

**Graph Condensation**    Graph condensation involves condensing knowledge from a large graph to create a smaller synthetic graph from scratch. However, the vanilla graph condensation often involves solving a expensive bi-level optimization problem Jin et al. (2021). Jin et al. (2022) further reduces the cost of graph condensation through one step gradient matching. We note that the graph condensation is orthogonal to our proposed method, as the final condensed graph still have the expensive `SpMM` operations.

## G  LIMITATIONS

Although our proposed method can effectively reduce the training time by reducing the number of active columns and rows for performing `SpMM` , it cannot directly reduce the memory usage for storing the large graph, which is another major bottleneck for scaling GNNs onto large graphs. When the memory is the major bottleneck, we recommend using our method jointly with other graph reduction methods ,e,g., graph sparsification Liu et al. (2023b).

## H  IMPACT STATEMENTS

This paper introduces research aimed at pushing the boundaries of Machine Learning. While our work might have several potential societal consequences, we feel there is nothing specifically to highlight here. You may include other additional sections here.

