# OpenReview forum: "STRUCTDROP: A STRUCTURED RANDOM ALGORITHM TOWARDS EFFICIENT LARGE-SCALE GRAPH TRAINING"
_ICLR.cc/2025/Conference — Submitted to ICLR 2025_

### Official Review · Reviewer_V73n · 2024-11-01

**Soundness:** 3
**Presentation:** 4
**Contribution:** 3
**Rating:** 8
**Confidence:** 4

**Summary:**

This paper presents an alternative way of speeding up SpMM during GNN training. The SOTA method (top-k sampling) involves picking row-column pairs that have the highest norm product. The interesting finding is that this tends to select a substantially similar subset of pairs in consecutive epochs and thus lead to under-fitting and lower accuracy. The proposed solution uses random sampling which shows good accuracy as well as speedup due to reduced workload in SpMM.

**Strengths:**

The finding of the reason Top-k sampling leads to lower accuracy is both intuitive and well supported by evidence.
The proposed solution is also intuitive and apparently effective.

**Weaknesses:**

N/A

**Questions:**

I am a little curious why wouldn't random sampling be one of the first things people try.

---

### Official Review · Reviewer_sWMT · 2024-11-03

**Soundness:** 1
**Presentation:** 2
**Contribution:** 1
**Rating:** 3
**Confidence:** 4

**Summary:**

The authors propose StructDrop, a random dropout technique for sparse matrix-matrix multiplication (SpMM) in both the forward and backward processes of Graph Neural Networks (GNNs). StructDrop applies instance normalization following each SpMM to mitigate the training shift due to random column-row pair dropping. Experimental results demonstrate that StructDrop achieves less training time with similar accuracies across different GNN architectures and GNN mini-batch training algorithms.

**Strengths:**

1. The proposed method is hardware-friendly and can achieve large speedup with negligible accuracy loss.

2. This paper introduces instance normalization to alleviate the distribution shift after sampling, which effectively maintains accuracy.

3. The proposed method provides significantly more acceleration with similar or better accuracies compared to previous baselines.

**Weaknesses:**

1. The proposed method combines dropping source nodes and instance normalization, which is relatively straightforward and may not significantly contribute to the GNN community. The justification of system-wise speed up and improving generalization is not sound because it seems a localized optimization and does not consider several systemic aspects in methodology and training (see below).

2. This method is limited to SpMM GNNs and cannot be applied to scatter-gather GNNs like GAT. Could the authors discuss the applicability of StructDrop to other GNN architectures beyond SpMM-based ones, and comment on potential ways to extend the approach to scatter-gather GNNs?

3. The paper claims that SpMM is the major bottleneck, consuming 70–90% of the total runtime. However, it overlooks cross-device data transfer as another bottleneck in mini-batch training on large-scale graphs where a single GPU cannot store the entire training data. Consequently, the proposed technique might not achieve significant speedup in these scenarios. Could the authors discuss how StructDrop would perform in scenarios where cross-device data transfer becomes a significant bottleneck, such as in mini-batch training on very large graphs? Are there ways the method could be adapted or combined with other techniques to address this issue?

4. The claim that DropNode and DropEdge operations are bottlenecks and that replacing them with StructDrop can achieve more than 2 times speedup is questionable. A runtime analysis of these operations with GPU implementations by DGL is necessary. The authors should compare the runtime of DropNode/DropEdge to SpMM under varying sparsity. Moreover, even if these runtimes are significant, the latencies of DropNode/DropEdge can be easily hidden as they are independent of GNN training. Could the authors provide a detailed runtime analysis comparing StructDrop, DropNode, and DropEdge operations using GPU implementations (e.g., from DGL), including comparisons of their runtimes to SpMM under varying sparsity levels? Additionally, could they discuss how the potential for hiding DropNode/DropEdge latencies impacts the overall speedup claims?

5. The baselines used in this paper are weak in terms of data augmentation and training acceleration. Stronger baselines are needed for a more comprehensive comparison.

6. A wider range of large-scale datasets with diverse statistics is required. Current results indicate that the speedup is highly correlated with graph density, with StructDrop achieving significant speedup only on datasets with substantially large average degrees. A thorough discussion of the work's limitations is necessary. Could the authors include experiments on additional large-scale datasets with varying graph densities and other properties? Additionally, could they provide a more comprehensive discussion of how graph properties impact StructDrop's performance, and what the limitations of the approach are for different types of graphs?

**Questions:**

1. The paper's main contribution is training acceleration. However, unlike top-K sampling, which benefits from a high cache hit ratio, uniform sampling only reduces FLOPs, which is insufficient. The authors should explore more advanced sparsification techniques that better leverage hardware properties, such as the memory hierarchy.

2. The analysis of how distribution shift occurs and how instance normalization mitigates this issue lacks clarity. Additionally, the authors should explain why they chose instance normalization over layer normalization.

3. A more comprehensive analysis of how various graph dropout techniques impact training and generation (Appendix E) would be beneficial.

4. Please also address the questions in the weakness section.

---

### Official Review · Reviewer_xaAg · 2024-11-03

**Soundness:** 2
**Presentation:** 3
**Contribution:** 2
**Rating:** 3
**Confidence:** 3

**Summary:**

The paper proposes a sampling method to accelerate the neighbor aggregation of graph neural network. The main observation made by the authors is that importance sampling leads to the sampling of same column-row pairs across training iterations. The authors proposed uniform sampling to overcome the problem and show better performance compared to importance sampling.

**Strengths:**

1. The paper is well-written. The proposed technique is simple and clear.

**Weaknesses:**

1. Limited novelty. The proposed sampling is very similar to the well-known layer-wise sampling technique for GNNs [Huang et al. 2018, Zou et al. 2019]. The sampling of the adjacency matrix rows corresponds to the sampling of neighboring nodes in a layer. While the authors claim that the proposed sampling technique can be "seamlessly combined with previous sampling methods", the difference is unclear to me. In fact, I feel that the proposed technique can be precisely expressed within the previous layer-wiser sampling framework.


2. The experiments are insufficient in terms of GNN models and data graphs:
- The authors evaluated their techniques with GCNs. Is the proposed technique applicable to attention-based models?
- The graphs used are small. It will be more convincing to evaluate on larger graphs where sampling is indeed beneficial.

3. Lacks technical depth. Sampling column-row pairs to speed up matrix multiplication is a well-known technique. It seems the main contribution of this paper is the experimental observation that importance sampling leads to under-fitting, and naive uniform sampling performs better in practice. The paper will be stronger if the authors can provide some theoretical insight.

**Questions:**

1. Can you clarify the difference between the proposed method and previous layer-wise sampling methods?

---

### Official Review · Reviewer_3wSj · 2024-11-04

**Soundness:** 3
**Presentation:** 3
**Contribution:** 2
**Rating:** 3
**Confidence:** 3

**Summary:**

The paper introduces StructDrop, a structured random sampling algorithm aimed at improving the efficiency of training Graph Neural Networks on large-scale graphs. Traditional GNN training is computationally intensive due to the message-passing mechanism, particularly the SpMM. Prior methods like top-k sampling, DropEdge, and DropNode attempt to reduce computational costs but often suffer from inefficiencies due to the overhead of reconstructing sparse matrices and can lead to underfitting.

StructDrop proposes to address these issues by uniformly sampling and removing entire columns (and their corresponding rows) from the sparse adjacency matrix, effectively reducing the computational complexity of SpMM without the need for costly sparse matrix reconstruction. To mitigate the variance and distribution shift introduced by random sampling, the authors incorporate instance normalization after the approximated SpMM operations. The method aims to balance computational efficiency with model performance. The results suggest that StructDrop can achieve up to 5.29× end-to-end speedup with a similar accuracy compared to standard GNN training.

**Strengths:**

Simplicity of Implementation: The proposed method is straightforward to implement, involving uniform sampling of columns and rows in the adjacency matrix and the application of instance normalization.

Empirical Performance: The experimental results show that StructDrop can achieve significant speedups in training time while maintaining comparable accuracy to baseline methods on several datasets and GNN architectures.

Practical Motivation: The paper addresses a practical problem in training efficiency for large-scale GNNs, which is of interest to the research community and industry practitioners dealing with big graph data.

**Weaknesses:**

Lack of Novelty: The method appears to be a combination of existing techniques—specifically, dropping nodes (similar to DropNode) and applying instance normalization. The paper does not sufficiently differentiate StructDrop from these prior methods in terms of novelty.

Insufficient Theoretical Justification: There is a lack of theoretical analysis explaining why uniform sampling combined with instance normalization effectively preserves model accuracy while reducing computational cost. The paper would benefit from theoretical insights or proofs to support the empirical findings.

Baselines: The experimental comparisons are primarily against older methods like DropEdge and DropNode. The paper does not compare StructDrop with more recent or advanced methods for efficient GNN training, such as graph sparsification techniques, quantization methods, or other modern sampling strategies.

Limited Analysis of Instance Normalization: The role of instance normalization in mitigating the effects of random sampling is not thoroughly analyzed. The paper lacks detailed experiments or theoretical explanations demonstrating why instance normalization is essential in this context.

Questionable Acceleration Claims: The claimed acceleration may not be as significant in practice because the latency reduction from the proposed method could be overshadowed by other bottlenecks in GNN training. Additionally, the paper does not discuss whether the latency improvements are due to algorithmic efficiency or simply hardware optimizations that might not generalize across different environments.

Missing Discussion on Limitations: The paper does not explore potential limitations of StructDrop, such as its performance on extremely large graphs, its impact on memory usage, or scenarios where the method might not provide significant benefits.

**Questions:**

Novelty Clarification: Can the authors clarify how StructDrop differs fundamentally from existing methods like DropNode combined with instance normalization? What are the unique contributions that set this work apart?

Theoretical Analysis: Is there a theoretical basis for why uniform sampling of columns and rows, along with instance normalization, maintains model performance? Providing theoretical justification or proofs would strengthen the validity of the approach.

Comparison with Other Baselines: Why were more recent methods for efficient GNN training not included in the comparisons? For instance, methods involving quantization, advanced graph sparsification, or other sampling techniques. Including these would provide a better context for evaluating StructDrop's effectiveness.

Impact of Instance Normalization: Could the authors provide a deeper analysis of the role of instance normalization? Specifically, how does it mitigate the variance introduced by random sampling, and what is its impact on training dynamics and final model performance?

Applicability to Other GNN Models: Have the authors tested StructDrop on attention-based GNNs or other architectures with different message-passing schemes? If not, what challenges do they anticipate in applying StructDrop to these models?

Guidelines for Sampling Ratio: Is there an optimal range for the sampling ratio that balances efficiency and accuracy? How sensitive is the method to this hyperparameter, and how should practitioners choose it in different scenarios?


While the paper addresses an important problem in GNN training efficiency, the current form lacks sufficient novelty and theoretical grounding. The method seems to be an incremental improvement over existing techniques without providing significant new insights. To enhance the contribution, the authors should:
- Strengthen the Theoretical Foundation: Provide theoretical analyses or proofs explaining why the proposed method works and under what conditions it is effective.
- Compare with Stronger Baselines: Include comparisons with more recent and relevant methods in efficient GNN training to demonstrate the advantages of StructDrop convincingly.
- Deepen the Analysis of Instance Normalization: Offer a detailed exploration of how instance normalization contributes to the method's success, possibly with ablation studies or theoretical explanations.
- Discuss Limitations and Applicability: Provide a balanced discussion of the method's limitations and applicability to a broader range of GNN architectures.
- Provide Implementation Details: Include more information on hyperparameters, implementation specifics, and possibly share code to enhance reproducibility.

By addressing these points, the paper would offer a more substantial contribution to the field and better meet the standards of a high-impact conference. I can increase my score to 5 based on the rebuttal.

---

### Meta-Review · Area_Chair_Sshy · 2024-12-25

**Metareview:**

This paper proposed structureDrop to speed up the GNN training. The main idea is to sample columns and rows from a sparse matrix for efficient SpMM computation.  An instance normalization is applied to mitigate the variance and distribution shift due to column/row dropping. However, the novelty of this approach is questionable, and the paper lacks a comparison with established baseline methods.

**Additional Comments On Reviewer Discussion:**

The main concerns from the reviewers are novelty of the paper and missing comparison with baselines. No rebuttal were provided by the authors.

---

### Decision · Program_Chairs · 2025-01-22

Reject